# Probiotic *Limosilactobacillus reuteri* DSM 17938 Changes Foxp3 Deficiency-Induced Dyslipidemia and Chronic Hepatitis in Mice

**DOI:** 10.3390/nu16040511

**Published:** 2024-02-12

**Authors:** Erini Nessim Kostandy, Ji Ho Suh, Xiangjun Tian, Beanna Okeugo, Erin Rubin, Sara Shirai, Meng Luo, Christopher M. Taylor, Kang Ho Kim, J. Marc Rhoads, Yuying Liu

**Affiliations:** 1Department of Pediatrics, Division of Gastroenterology, McGovern Medical School, The University of Texas Health Science Center at Houston, Houston, TX 77030, USA; erini.kostandy@gmail.com (E.N.K.); beanna.okeugo@uth.tmc.edu (B.O.); 2Department of Anesthesiology, Critical Care and Pain Medicine, McGovern Medical School, The University of Texas Health Science Center at Houston, Houston, TX 77030, USA; ji.ho.suh@uth.tmc.edu (J.H.S.); kangho.kim@uth.tmc.edu (K.H.K.); 3Department of Bioinformatics and Computational Biology, The University of Texas MD Anderson Center, Houston, TX 77030, USA; xiangjun.tian@gmail.com; 4Department of Pathology and Laboratory Medicine, McGovern Medical School, The University of Texas Health Science Center at Houston, Houston, TX 77030, USA; erin.rubin@uth.tmc.edu (E.R.); sara.shirai@uth.tmc.edu (S.S.); 5Department of Microbiology, Immunology and Parasitology, Louisiana State University Health Sciences Center, New Orleans, LA 70112, USA; mluo2@lsuhsc.edu (M.L.);

**Keywords:** probiotics, regulatory T cell, autoimmunity, lipid metabolism, microbiota, inflammation, IPEX syndrome, autoimmune disease, scurfy mouse

## Abstract

The probiotic *Limosilactobacillus reuteri* DSM 17938 produces anti-inflammatory effects in scurfy (SF) mice, a model characterized by immune dysregulation, polyendocrinopathy, enteropathy, and X-linked inheritance (called IPEX syndrome in humans), caused by regulatory T cell (Treg) deficiency and is due to a Foxp3 gene mutation. Considering the pivotal role of lipids in autoimmune inflammatory processes, we investigated alterations in the relative abundance of lipid profiles in SF mice (± treatment with DSM 17938) compared to normal WT mice. We also examined the correlation between plasma lipids and gut microbiota and circulating inflammatory markers. We noted a significant upregulation of plasma lipids associated with autoimmune disease in SF mice, many of which were downregulated by DSM 17938. The upregulated lipids in SF mice demonstrated a significant correlation with gut bacteria known to be implicated in the pathogenesis of various autoimmune diseases. Chronic hepatitis in SF livers responded to DSM 17938 treatment with a reduction in hepatic inflammation. Altered gene expression associated with lipid metabolism and the positive correlation between lipids and inflammatory cytokines together suggest that autoimmunity leads to dyslipidemia with impaired fatty acid oxidation in SF mice. Probiotics are presumed to contribute to the reduction of lipids by reducing inflammatory pathways.

## 1. Introduction

Regulatory T (Treg) cells are crucial for maintaining peripheral tolerance and inflammatory T cell suppression [1,2]. The Forkhead Box protein 3 (Foxp3) gene is a master transcription factor involved in Treg cell development, stability, and function [3,4]. In mice, Foxp3 gene mutation results in the scurfy (SF) mouse model, which serves as a unique model for a rapidly fatal disease characterized by immune dysregulation, polyendocrinopathy, enteropathy, and X-linked inheritance (human IPEX syndrome [5,6,7]. IPEX syndrome is linked to various autoimmune disorders including type I diabetes (T1DM), eczema, thyroid dysfunction, interstitial pneumonitis, and renal disease [8]. In addition, there have been case reports of autoimmune hepatitis (AIH) in patients with IPEX syndrome [9].

SF mice develop a severe autoimmune phenotype mediated by uncontrolled Th1 and Th2 cells, resulting in multi-organ failure and premature death before four weeks of age [10,11]. The affected organs include the lungs (pneumonitis), skin (severe dermatitis and autoimmune blistering disease), joints (arthritis), kidneys (glomerulonephritis), and reproductive organs [2,12,13,14]. Additionally, SF mice demonstrate hematologic abnormalities and display phenotypic features like autoimmune systemic lupus erythematous (SLE), including the presence of anti-nuclear antibodies, anti-double-stranded DNA antibodies, anti-histone antibodies, and anti-Smith antibodies [15]. In addition, they exhibit histological and biochemical features of AIH and autoimmune cholangitis [16,17]. 

We previously demonstrated dynamic changes in autoimmunity and gut microbial dysbiosis during the lifespan of SF mice. We were able to modify these changes through intragastric administration of probiotic *Limosilactobacillus reuteri* DSM 17938 (DSM 17938), resulting in marked prolongation of lifespans from <1 month to >4 months [18]. DSM 17938, derived from ATCC 55730, was isolated from a Peruvian mother’s breast milk by removing two plasmids harboring antibiotic-resistance genes [19]. This modified strain has been shown to be clinically beneficial in newborn conditions such as infantile colic [20,21,22] and necrotizing enterocolitis (NEC) [23]. In rodents with experimental NEC, immune deficiency, and healthy newborns, this strain has been observed to reset gut microbial dysbiosis, generate beneficial metabolites, and regulate immune responses [18,24,25,26]. In SF mice, we identified a unique anti-inflammatory pathway which operates through an adenosine/inosine-A_2A_-dependent mechanism, resulting in a profound reduction of inflammatory T cells [18,27]. 

The development of autoimmune disorders is influenced by metabolic disturbances, particularly those related to lipid metabolism. These abnormalities of lipid metabolism are notable in both polygenic autoimmune disorders such as T1DM [28] and in monogenic primary autoimmune diseases such as common variable immunodeficiency (CVID) [29]. Lipids are known to play crucial roles in inflammatory processes [30,31]. For example, some polyunsaturated fatty acids have been shown to have anti-inflammatory effects in autoimmune disorders such as SLE and T1DM, while others are pro-inflammatory [32]. In addition, the gut microbiome plays an important role in regulating intestinal lipid metabolism in both human and animal models [33,34]. Gut microbial dysbiosis may contribute to dysregulating lipid metabolism in primary immune deficiency [35,36] and other diseases such as inflammatory bowel disease (IBD), thereby highlighting the importance of this association and its potential benefit as a therapeutic target [33,37]. 

The changes in circulating lipids and the association of lipids with gut microbiota and systemic inflammation in SF mice have not been investigated. We hypothesized that Foxp3 deficiency may be associated with the dysregulation of lipid metabolism. We previously demonstrated dynamic gut microbial dysbiosis throughout the first 21 days of life of SF mice. This dysbiosis was beneficially modulated through the intragastric administration of probiotic DSM 17938 to SF mice [18]. Consequently, we hypothesized that DSM 17938 could alter lipid derangements if it is present. In the current study, we examined plasma lipid profiles and hepatic inflammation in SF mice, comparing the changes to normal mice and SF mice treated with DSM 17938. Our aim was to investigate the potential effect of DSM 17938 on these changes. Additionally, we explored correlations between lipids and gut microbes, circulating inflammatory biomarkers, and liver genes involved in lipid metabolism to further understand the probiotic mechanism of action in Treg-deficient autoimmunity. 

## 2. Materials and Methods

*Mice.* Wild-type (WT) C57BL/6J (000664) male and heterozygous B6.Cg-Foxp3sf/J (004088) female mice, 6–8 weeks old, were purchased from the Jackson Laboratory (Bar Harbor, ME) and were allowed to acclimatize for 2 weeks before setting up breeding pairs for generating SF mice (B6.Cg-Foxp3sf/Y). Since the Foxp3 gene is located on the X chromosome, only males had SF features, with a 25% probability of total offspring from each litter being SF mice. SF mice were collected from at least 3 different cages per experimental group, and only male mice (either SF or WT littermates) were used in this study. 

SF mice exhibited scaly skin on their ears, eyes, and tails and had deformed ears beginning on day of life 13, and early deaths were noted around d24–28 of life [18,38]. Male mice were treated with either control medium or the probiotic beginning on d8 of life, prior to clinical recognition, and were analyzed on d21 of life at the weaning date. The mice were housed under a 12 h light/12 h dark cycle and temperatures of 18–23 °C with 40–60% humidity. They had access to food and water ad libitum in a specific-pathogen-free (SPF) animal facility at the University of Texas Health Science Center at Houston (UTHealth). This study was carried out in accordance with the recommendations of the Guide for the Care and Use of Laboratory Animals of the National Institutes of Health (NIH). The Institutional Animal Care and Use Committee (IACUC) of UTHealth approved the study (protocol numbers: AWC-14-056, AWC-17-0045, and AWC-22-0112). 

*Preparation of DSM 17938 and treatment of mice.* DSM 17938, provided by BioGaia AB (Stockholm, Sweden), was prepared as described previously [39]. Briefly, DSM 17938 was anaerobically cultured in deMan-Rogosa-Sharpe (MRS) medium at 37 °C for 24 h, and then plated in MRS agar at specific serial dilutions and grown anaerobically at 37 °C for 48–72 h. A quantitative analysis of bacteria in culture media was performed by comparing optical density (OD) 600 nm of cultures at known concentrations using a standard curve of bacterial colony-forming units (CFU)/mL grown on MRS agar. Freshly cultured DSM 17938 bacteria were re-suspended in specified volumes of fresh MRS media based on the calculated CFU required for each feeding, prepared daily for mouse feeding. 

Newborn mice were fed with DSM 17938 (10^7^ CFU/day in 100 µL) using intragastric administration, daily, starting from day of life 8 (d8) to d21 (SFL, n = 5) or compared with SF mice (SFC, n = 6) and WT male littermate controls (WTC, n = 6) that were fed with an identical volume of fresh MRS media in the absence of DSM 17938. The dosage and chosen administration method were based on documented evidence of the probiotic strain’s efficacy. For infantile colic, beneficial effects were observed with oral administration of ~5 × 10^8^ CFU (5 drops) in sunflower oil to the babies [22,40,41]. In neonatal experimental NEC in rodents, a daily intragastric administration of 10^6^ cfu/g. b.w./day beginning on d10 had been established [24,39,42]. In Treg-deficient SF mice, daily intragastric administration of 10^7^ cfu/day until d21 reduced the severity [18,25,43]. Mice were euthanized at age d22 to collect blood and cecal/colonic contents. The isolated plasma and cecal contents were stored immediately at −80 °C for further plasma lipid profile and fecal microbiota analysis. 

*Plasma global lipid profile analysis.* Plasma lipid metabolites were processed and assayed using Metabolon Inc. “www.metabolon.com (accessed on 6 February 2024)” [18]. A total of 212 named lipids in plasma were detected using a non-targeted metabolomic analysis platform including ultra-high-performance liquid chromatography/electrospray ionization tandem mass spectrometry (UPLC-MS/MS) and gas chromatography/mass spectrometry (GC/MS). The lipid profile data included fold changes of SFC/WTC (SF vs. Control) and SFL/SFC (SF+ DSM 17938 treatment vs. SF + MRS treatment) and were reported by Metabolon Inc., with *p* < 0.05 indicating a significant difference between the groups using the Welch’s two-sample *t*-test.

*Stool microbial community analysis.* Sequencing and bioinformatics were performed at Louisiana State University Health Sciences Center Microbial Genomics Resource Group “http://metagenomics.lsuhsc.edu/mgrg (accessed on 6 February 2024)”. The 16S ribosomal DNA hypervariable region V4 was PCR-amplified using primers V4F GTGCCAGCMGCCGCGGTAA and V4R GGACTACHVGGGTWTCTAAT with Illumina adaptors and molecular barcodes to produce amplicons. Samples were sequenced on an Illumina MiSeq (Illumina, San Diego, CA, USA) using a 500 cycle V2 sequencing kit to produce 2 × 250 paired end reads. The forward and reverse-read files were processed using the DADA2 [44] and pipelined in QIIME2 [45]. Amplicon sequence variants were taxonomically classified using the SILVA v138 database [46]. Bacterial alpha and beta diversity metrics, as well as taxonomic community assessments were performed using QIIME2. 

*Plasma cytokines, alanine aminotransferase (ALT), and aspartate aminotransferase (AST) measurement.* Plasma cytokines IFN-γ and IL-4 were assessed using a mouse proinflammatory assay kit from Meso Scale Discovery (MSD), according to the manufacturer’s protocol. Plasma ALT and AST levels were measured using a Beckman Coulter AU480 Chemistry Analyzer using the Center for Comparative Medicine, Pathology Diagnostic Laboratory, Baylor College of Medicine, Houston, Texas, data reported as U/L.

*Histological evaluation of hepatitis in mouse livers.* Liver tissues collected from mice were fixed and processed using the Cellular and Molecular Morphology Core Laboratory at Texas Medical Center Digestive Diseases Center, Houston, Texas, and stained with hematoxylin and eosin (H & E) for histological evaluation. Hepatitis evaluation was performed by two pathologists independently, using the modified hepatic activity index (abbreviated as modified HAI) grading system. We scored periportal or periseptal interface hepatitis (0 to 4), confluent necrosis (0 to 6), focal lytic necrosis, apoptosis, and focal inflammation (0 to 4), portal inflammation (0 to 4), and modified staging (architectural changes, fibrosis, and cirrhosis) (0 to 6) [47]. 

*Genes associated with lipid metabolism analyzed using a quantitative real-time polymerase chain reaction (RT-qPCR).* Total RNA was extracted from mouse liver tissues using the RNAeasy Mini Kit (QIAGEN), according to the manufacturer’s protocol. The total RNA (500 ng) was reverse transcribed using amfiRivert cDNA synthesis Platinum Master Mix (GenDepot). qRT-PCR was performed using amfiSure qGreen Q-PCR Master Mix (GenDepot) on the CFX Opus 384 Real-Time PCR System (BIO-RAD). The peroxisome proliferator-activated receptor (*Ppara = Pparα*) and its target genes [48], genes involved in de novo lipogenesis [49] and genes that are involved in lipid and fatty acid uptake [50,51], were evaluated. All qPCR primers are listed in Appendix A.

*Statistical Analysis.* We measured the difference in individual lipid metabolites from the testing groups and reported fold changes; the significance of differences was tested using one-way ANOVA. The upregulated and downregulated lipid metabolites were defined as those with 2.0 (up) or 0.5 (down)-fold changes when associated with a *p*-value < 0.05. Integrative analysis of lipid metabolites and gut microbiota: lipid metabolites and plasma cytokine levels were measured by calculating the Spearman’s rank correlation coefficient using the matched samples of each group with both lipid metabolites and microbiota data or plasma cytokine data. A heatmap was plotted using the R package heatmap. For gene expression analysis, to compare the groups of SFC, SFL, and WTC, we used a two-way ANOVA with Tukey’s multiple comparisons. For histological parameters, to compare the groups of SFC, SFL, and WTC, we used one-way ANOVA with Tukey’s multiple comparisons test using GraphPad Prism version 9.4.1 (GraphPad Software, San Diego, CA, USA). Data are represented as means ± SD. *p* values < 0.05 were considered statistically significant. 

## 3. Results

### 3.1. Altered Plasma Lipid Profiles in Treg-Deficiency SF Mice

Altogether, 212 lipid metabolites were identified in the plasma of mice. The SFC group exhibited significant dyslipidemia in comparison to the WTC mice, as evidenced by the significant upregulation of 114 (54%) lipid metabolites, with only 9 (4%) lipids showing downregulation. Specifically, there was a notable increase in the levels of 14 phospholipids, 14 inositols, 12 acylcarnitines, 11 polyunsaturated fatty acids (PUFAs), 11 long-chain FAs (LCFAs), 11 mono- or di-acylglycerols, nine lysolipids, six sphingolipids, four FAs involved in branched-chain amino acid metabolism, three medium-chain FAs (MCFAs), and three acylglycines (Figure 1a). Among the individual upregulated lipids in the SFC group compared to the WTC group, monoacylglycerols and diacylglycerols exhibited the highest fold-change increases, exceeding four times the baseline levels in WTC mice (Figure 1b).

### 3.2. Probiotic DSM 17938 Modulates Plasma Lipid Profile in SF Mice

Intragastric administration of DSM 17938 to SF mice was associated with downregulation of 37 lipids (17%), including sub-pathways phospholipids, acylcarnitine, dihydroxy FA, monoacylglycerol, polyunsaturated FAs, BCAA, and acylglycine. Only four lipids (2%) were upregulated. 

Treatment with DSM 17938 resulted in the downregulation of many lipid pathways, including eight phospholipids (the same phospholipids upregulated in SF mice), four acylcarnitines, and MCFAs, three dicarboxylate FAs, and two of each of the following lipid categories, dihydroxy FAs, LCFAs, monoacylglycerol, branched FAs. Additionally, there was downregulation observed in one PUFA and one acylglycine (Figure 2).

### 3.3. Foxp3^+^ Treg Deficiency Reduces Expression of Genes That Are Related to Lipid Metabolism in the Liver of SF Mice

To investigate the mechanism of plasma dyslipidemia caused by Treg-deficiency, we analyzed genes associated with lipid metabolism in the liver of SF mice. PPARα is a ligand-activated transcription factor that belongs to the steroid hormone receptor superfamily, which is expressed predominantly in tissues that have a high level of fatty acid catabolism [52]. PPARα regulates the expression of several genes critical for lipid and lipoprotein metabolism [52]. Expression levels of Ppara and its targets, as well as genes related to fatty acid synthesis and uptake were downregulated (Figure 3). Notably, intragastric administration of the probiotic did not rescue the inhibition of gene expression levels in SF mice.

### 3.4. Altered Plasma Lipids Are Correlated with Gut Microbiota in SF Mice

We have discovered that intragastric administration of DSM 17938 to SF mice ameliorates Treg-associated gut microbial dysbiosis. We found that the decreased Shannon α-diversity associated with Treg deficiency was reversed with DSM 17938 treatment, and that a three-dimensional principal coordinate analysis (PCoA) revealed SF mice with DSM 17938 treatment displayed a shift in microbial community, which was distinct from either WT or SF populations [18,25]. An integrative analysis of changed plasma lipids and gut microbiota revealed a number of correlated alterations, as demonstrated using the HeatMap (Figure 4a). We found that altered lipid metabolites in 19 lipid subpathways significantly correlated with 12 genera of fecal bacteria (Figure 4b). Phospholipids, acylcarnitines, PUFAs, LCFAs, and acylglycerols were associated with several common genera of bacteria. All the upregulated lipids were positively associated with *Escherichia* and were negatively associated with *Ruminococcus*. Among them, phospholipids correlated with the highest number of bacteria, including seven different bacterial genera. There were 47 acylcarnitine−derivatives correlated with bacteria, showing positive correlation with *Bacteroides*, *Pseudomonas*, *Anaerotruncus*, and *Escherichia*, and negative correlation with *Ruminococcus*, *Anaeroplasma*, *Turicibacter*, and *Akkermansia*. As mentioned, 17 percent of plasma lipids were downregulated by DSM 17938, and these lipids correlated with *Escherichia*, *Pseudomonas*, and *Bacteroides*. 

### 3.5. Changed Plasma Lipids Are Positively Correlated with Plasma Th1-Associated (IFN-γ) and Th2-Associated (IL-4) Cytokine Levels

We have previously demonstrated significant increases in plasma Th1- and Th2-associated cytokines in SF mice, levels of which were reduced through intragastric administration of DSM 17938 [18]. Certain lipids have been identified as potential contributors to systemic inflammation in primary autoimmune diseases [29]. To understand which lipid metabolites were associated with Th1- and Th2-associated inflammation, we conducted a correlation analysis between upregulated lipid metabolites and plasma IFN-γ and IL-4. The results showed significant positive correlations between 1-linoleoyl-GPA, caprylate, stearoyl sphingomyelin, taurochenodeoxycholate, and aurohyodeoxycholic acid with IFN-γ (Table 1). Additionally, three lysolipids, 1-palmitoyl-2-oleoyl-GPC, LCFAs (eicosenoate and erucate), PUFA (adrenate), and 1-(1-enyl-palmitoyl)-GPE were positively correlated with IL-4 (Table 2).

### 3.6. Foxp3^+^Treg Deficiency Induces Chronic Hepatitis and Can Be Partially Reduced through Intragastric Administration of DSM 17938

Although SF mice had significantly upregulated lipids in plasma, hepatic steatosis was not observed in histological evaluation. Instead, the livers of SF mice demonstrated heavy portal tract and periportal chronic inflammation with interface hepatitis, and central and portal vein endothelialitis. With intrasgastric administration of DSM 17938, SF mice showed marked reduction of the portal tract and subendothelial central vein lymphocytic inflammation (Figure 5a). The average total modified HAI score for SF mice was 5.45 ± 1.48 (of a maximal total score of 24), whereas normal WT mice scored 0 (no changes associated with autoimmunity) (Figure 5b). SF mice scored a mean of 1.95 ± 0.76 in periportal or periseptal interface hepatitis (Figure 5c); 0 in confluent necrosis; 1.25 ± 0.42 in focal lytic necrosis, apoptosis, and focal inflammation (Figure 5d); 2.25 ± 0.67 in portal inflammation (Figure 5e); and 0 in modified staging (no observed architectural changes, fibrosis, or cirrhosis). Interestingly, venous endothelial inflammation involving most of the portal and hepatic venules was much like that seen in AIH and acute T-cell mediated rejection of the liver in human patients. DSM 17938 impacted the total score, periportal or periseptal interface hepatitis, and portal inflammation scores (Figure 5b,c,e), while no improvement was noted in the focal inflammation score (Figure 5d). However, elevated plasma levels of ALT and AST were not observed in SF mice compared to WT mice (Appendix A).

## 4. Discussion

We have uncovered the presence of dyslipidemia in Treg-deficiency-induced autoimmunity. Altered plasma lipids were significantly correlated with systemic inflammation and with specific genera of gut bacteria. Treg-deficiency due to Foxp3 gene mutation resulted in the histological features of hepatitis which could be ameliorated using gavage feeding DSM 17938. Our study highlights the interplay of liver gene expression, immune cell cytokines, gut microbes, and microbe-associated metabolites in this autoimmune disease. 

### 4.1. Altered Lipid Metabolites in Mice and Humans with Treg-Deficiency Are Associated with Inflammation

Bioactive lipid metabolites play crucial roles in inflammatory processes and regulate immune responses while influencing leukocyte trafficking and clearance in autoimmune diseases [30,53]. Our observations show significant lipid derangements in SF mice, with a >50% elevation of identified lipid metabolites including PUFAs, acylcarnitines, acylglycines (acylglycerols), monoacylglycerol compounds, sphingolipids, and LCFAs. Upregulated lipids are also found in patients with autoimmune conditions such as T1DM, rheumatoid arthritis (RA), multiple sclerosis (MS), SLE, and IBD [54]. In human IPEX syndrome, endocrinopathy evolves during the first year, while T1DM usually develops in the first month of life due to extreme autoimmunological reactions from activated T cells [55]. However, there have been no studies of lipidomics in human IPEX syndrome or in SF mice. In our previous studies, we showed a dynamic progression of autoimmunity development over the first 22d of life. We demonstrated increased levels of IFN-γ and IL-4 in the plasma along with IFN-γ- and IL-4- producing CD4^+^T cells in the spleen. High cytokines were detectable as early as d8 of age, even before the manifestation of clinical symptoms. Notably, these elevated levels of Th1- and Th2-associated cytokines persisted throughout d22 of age [18]. In the current study, we found that the upregulation of lipids positively correlated with circulating IFN-γ and IL-4 levels.

### 4.2. Categories of Lipids That Are Increased in Autoimmune Diseases

High serum PUFAs, including ω-3 and ω-6, are observed in SLE and correlate with elevated anti-nuclear antibody (ANA) titers, and levels responded to immunosuppressants [56]. Notably, our SF mice also displayed significant elevations in the same PUFAs seen in human SLE, including docosahexaenoic (DHA), eicosapentaenoic acid (EPA), stearidonic, linolenic, and arachidonic acid. Some studies have demonstrated that ω-3 PUFAs have anti-inflammatory functions and could have therapeutic potential in autoimmune diseases [57]. Therefore, it is plausible to hypothesize that the increased PUFAs in SF mice may represent a response to inflammation.

Acylcarnitines, involved in LCFA transportation into the mitochondria, are increased in several autoimmune diseases, and have been used as important diagnostic markers for inborn errors of fatty acid oxidation, as well as markers for energy metabolism, deficits in mitochondrial and peroxisomal β-oxidation activity, insulin resistance and physical activity [58]. Recently they have been identified as potential biomarkers for the diagnosis of SLE [59]. Given the presence of similarities between SF mice and SLE [15], we may postulate that acylcarnitines may also be useful biomarkers in Treg-associated autoimmune disorders [35]. 

Monoacylglycerol compounds were significantly upregulated in the plasma of patients with systemic sclerosis (SSc). These may function as endogenous cannabinoid ligands involved in SSc pathogenesis [60]. The elevation of sphingolipids, including ceramide, described in SLE and RA, can lead to apoptosis, endothelial dysfunction, and perpetuation of autoimmunity [61]; hence, a similar mechanism may be postulated in Treg-associated autoimmune pathogenesis. It is noteworthy that a diet rich in LCFAs can contribute to nonalcoholic steatohepatitis development in human and guinea pig models [62]. Despite elevated LCFAs in SF mice, liver histology did not show signs of steatosis, indicating that LCFAs may contribute to hepatitis by exaggerating T helper immune responses rather than causing steatohepatitis.

LCFAs, particularly, very-long-chain fatty acids (VLCFAs), which are abundant in myelin, have been implicated in autoimmune-mediated neuroinflammation, including the development of MS, with evidence of elevated sphingosine-1-phosphate (S1P) levels in glia. Therapeutic effects of S1P inhibition in a mouse model of MS has been reported [63]. Interestingly, it has been previously shown that DSM 17938 reduced the severity of autoimmune encephalomyelitis in a mouse model with MS [64]. 

### 4.3. Potential Role of PPARα in Dyslipidemia and Chronic Hepatitis in SF Mice

The reduced expression levels of genes associated with de novo lipogenesis, lipid and fatty acid uptake, and fatty acid oxidation in SF mice, may be secondary to inflammation in SF mice. The PPARα pathway is primarily expressed in rodent hepatocytes and is responsible for fat metabolism and carbohydrate homeostasis, as well as cell proliferation and differentiation and inflammation. The roles of PPARs and their receptors in chronic diseases such as diabetes, cancer, and atherosclerosis are well established [65]. PPARα is a key regulator of fatty acid oxidation, and its activation leads to a decrease in lipid levels and elimination of triglyceride from plasma [66]. PPARα expression was downregulated in SF mice, along with its target genes, which would inhibit fatty acid oxidation and, subsequently, potentially upregulate lipid levels in the plasma [52,67]. Additionally, PPARα activation represses NF-κB signaling, resulting in decreased inflammatory cytokine production by different cell types, with reduced tumor necrosis factor-alpha (TNF-α), IL-6, and Il-1β [68]. PPARα inhibits activator protein −1 (AP-1)-dependent genes involved in inflammation and tumor progression [69] and suppresses Th17 cells through modulation of IL-6/STAT3/RORγt signaling in rat models of autoimmune myocarditis [70]. The reduction in PPARα activity in SF mice, coupled with dyslipidemia, may, therefore, be contributing to worsening autoimmunity, inflammation, and hepatitis. PPARα could potentially serve as a molecular target for the treatment of autoimmune diseases. 

Finally, gut bacteria and microbial-associated metabolites, such as bioactive lipids and endocannabinoids could serve as a PPARα agonist with potential benefits in human diseases [37]. For instance, *Akkermensia muciniphila* can activate PPARα via modulating endocannabinoid-related lipids, specifically, mono-palmitoyl-glycerols [71]. We, therefore, postulate that changes in the SF mice microbiome, including down regulation of *Akkermensia* species, might be associated with PPARα downregulation. However, further studies are necessary to validate this hypothesis. 

### 4.4. Potential Clinical Relevance of SF Mouse Chronic Hepatitis Phenotype

Dysfunction or deficiency of Tregs has been linked to the onset and progression of AIH [72]. AIH- associated autoantibodies have been described in patients with IPEX syndrome [9,73]. A recent study reported the clinical, serological, and immunopathological characteristics of AIH with primary biliary cholangitis (PBC) in SF mice [17]. Additionally, central perivenulitis (CP) was observed in SF mice livers, shedding light on the potential use of SF mice as an acute T-cell mediated rejection model in a transplant setting or pharmacologic therapy for human patients with AIH. Indeed, the pathophysiology of SF mice is mainly mediated by Th1 and Th2 cells and their associated cytokines [10,18,74]. 

### 4.5. DSM 17938 Impacts Inflammation and Autoimmune-Associated Lipids through Modulating Gut Microbiota and Microbial-Associated Metabolites in SF Mice

Gastrointestinal microbiota dysbiosis can contribute to autoimmune disorders [35,75,76]. Probiotics not only reshape the host microbiota but also impact global metabolic functions, offering potential autoimmune disorder treatments [77]. The gut microbiota influenced by DSM 17938 in SF mice has been associated with metabolic and immunomodulatory regulation [78]. The immunomodulatory mechanisms of DSM 17938 include promoting the maturation of tolerogenic dendritic cells (DCs) via a toll-like receptor (TLR2) and educating naïve T cell differentiation toward Treg cells [24]. Additionally, DSM 17938 directly inhibits inflammatory effector T cells [42] and inhibits the NF-kB pathway [39] reducing inflammatory cytokine production during inflammation. Probiotic-educated Treg cells maintain a regulatory function in neonatal stress conditions [79]. The primary mechanism through which Treg cells control inflammatory T cells is through the production of adenosine, which interacts with the adenosine receptor, specifically, A_2A._ This receptor is predominately present on inflammatory T cells and inhibits their differentiation [80]. 

In SF mice, the absence of this control mechanism could be reversed by probiotic-derived adenosine and the adenosine metabolite inosine. Adenosine and inosine act as agonists of the adenosine receptor A_2A_ to control inflammation [18,27,43]. DSM 17938 may influence other bacteria-associated metabolites, including amino acids and their derivatives such as glutamine, tryptophan/indoles, and polyamines. These metabolites can regulate the PPAR gamma (PPARγ) pathway (for example, glutamine), promote Treg cell activation (for example tryptophan-derived indoles), and facilitate renewal of the intestinal mucosa (for example, glutamine and polyamines) [26,81].

We found that DSM 17938 significantly alleviated hepatitis histology in SF mice. However, it only downregulated 17% of lipids, and could not rescue the downregulated genes associated with lipid metabolism. In SF mice, *Pseudomonads*, *Escherichia*, and *Bacteroides* positively correlated with upregulated lipids. These lipids were downregulated by DSM 17938. It is worth noting that the relative abundance (RA) of *Bacteroides* at the genus level in SF mice increased to 30%, which could be reduced to the normal levels of 15% by DSM 17938 [18]. *Bacteroides*, specifically *Bacteroides fragilis* containing the ubiquitin B (UBB) gene, may trigger autoimmune responses due to molecular mimicry in T1DM, IBD, and MS [82]. *P. aeruginosa*, a strain belonging to *Pseudomonads* can cause diseases across diverse host organisms [83]. 

Conversely, we found that the bacteria *Anaeroplasma* and *Ruminococcus* were negatively correlated with altered lipids in SF mice. These bacteria are generally considered beneficial to protect against autoimmunity. *Anaeroplasma* is stimulated by probiotic *Lactobacillus rhamnosus* GG in rodent models and may have the potential to produce anti-tumor effects [84]. In the setting of a high-fat diet, *Anaeroplasma* may contribute with other commensals to reduce the formation of atherosclerotic plaque [85]. Downregulation of *Ruminococcus* has been also associated with other autoimmune diseases [86]. SF mice showed a marginal reduction in *Ruminococcus* compared to WT mice, while intragastric administration of DSM 17938 to SF mice resulted in recovery of *Ruminococcus* to a level similar level to that in WT mice [18]. 

Finally, probiotics and gut commensals may cooperate in regulating enterocyte lipid metabolism. For example, L-lactate secreted from *Lactobacillus paracasei* promotes lipid storage, while acetate secreted from *Escherichia coli* stimulates lipid oxidation and consumption in enterocytes [34]. Therefore, exploring the dysfunction of lipid metabolism in SF mice’s enterocytes and gut microbial modulation of these processes holds promise for identifying therapeutic targets in these disorders. 

### 4.6. Study Limitations

Our data were analyzed on the weaning day (d22) of SF mice. The illness in SF mice leads to malnutrition because they consume less of the dam’s breast milk compared to their WT littermates, and malnutrition may also be a factor affecting the lipid metabolic profile. Also, in human AIH, serum enzyme ALT and AST levels are markedly increased [87]. However, we chose to study mice at less than 1 month of life because of severe multiorgan (including pulmonary) failure, ensuring their survival at the time of testing. These mice were on breast milk, as we intentionally avoided the effects of solid food on gut microbiota in this study. Therefore, we did not analyze liver injury beyond d22 of age. Liver cell death was relatively mild, even though we found lymphocytic infiltration in the face of normal plasma ALT and AST levels. 

## 5. Conclusions

Treg deficiency not only leads to systemic inflammation and malnutrition, but also results in a global increase in plasma lipids with decreased expression of hepatic genes associated with lipid transport, fatty acid oxidation, and de novo lipogenesis. Certain microbial taxa were linked to abnormal lipids, and these lipids were tightly associated with plasma IFN-γ and IL-4 levels. We found that administration of a single probiotic, *L. reuteri* DSM 17938, correlated with shifts in microbial taxa known to improve gut microbial carbohydrate and fat metabolism. *L. reuteri* improved liver periportal infiltration and endotheliitis. Further research on lipid metabolism and its connection with bacterial communities in SF mice may provide diagnostic biomarkers and therapeutic targets for treating autoimmune or transplant-associated liver disease.

## Figures and Tables

**Figure 1 nutrients-16-00511-f001:**
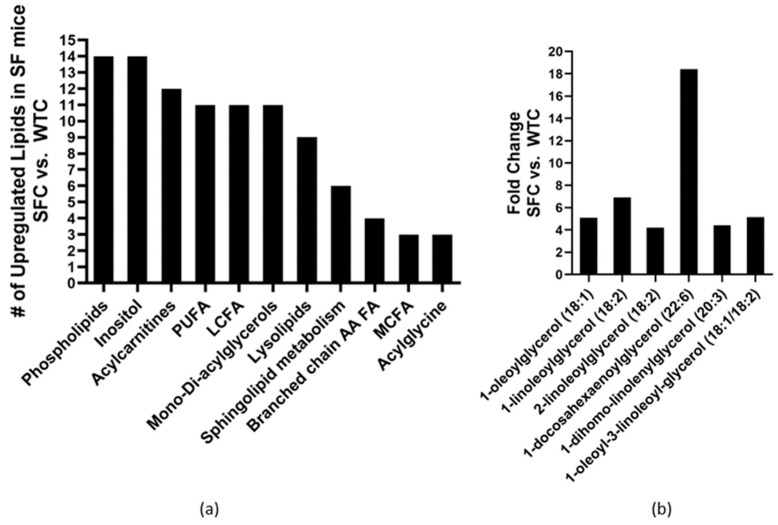
Changed plasma lipid profiles in Treg-deficiency SF mice: (**a**) the number of lipid sub-pathway categories were upregulated, and (**b**) Mono- and di-acylglycerols had >4-fold increase in SF mice (SFC, n = 6) compared to WT mice both fed with control media (WTC, n = 6).

**Figure 2 nutrients-16-00511-f002:**
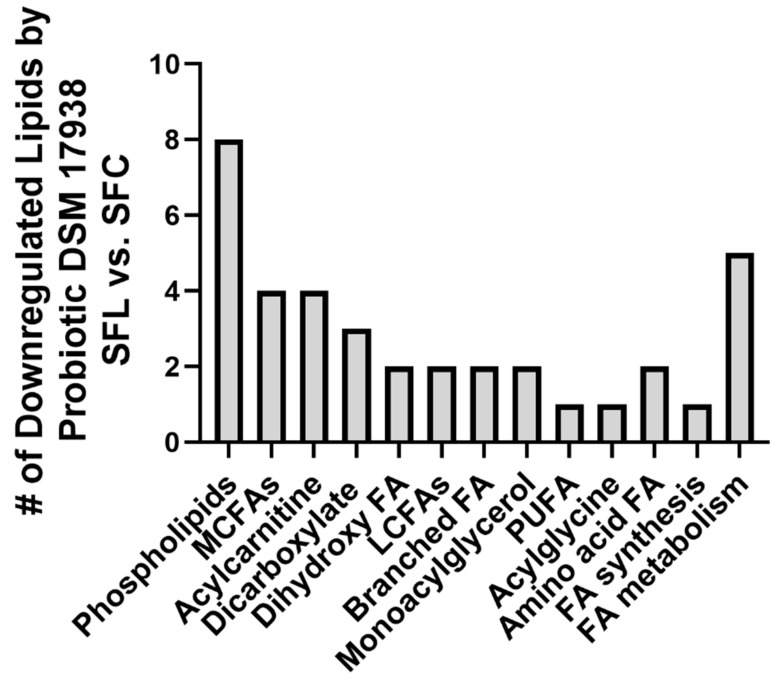
Downregulated plasma lipids in SF mice fed with probiotic DSM 17938. The number of downregulated lipids in SF mice fed with DSM 17938 (SFL, n = 5) was compared to SF mice fed with control media (SFC, n = 6).

**Figure 3 nutrients-16-00511-f003:**
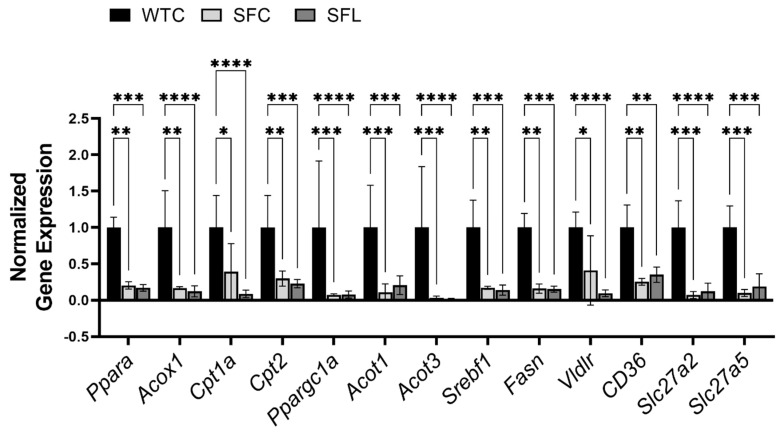
Foxp3 gene mutation globally and it severely reduced the expression levels of genes that are related to lipid metabolism in the liver of SF mice. *Pparα* and its target genes include acyl-CoA Oxidase 1 (Acox1), carnitine palmitoyltransferase (Cpt1a and Cpt2), peroxisome proliferator-activated receptor gamma coactivator 1-alpha (Ppargc1a), and acyl-CoA thioesterase (Acot1, and Acot3) [48]. Genes involved in de novo lipogenesis [49] include sterol regulatory element binding transcription factor 1 (*Srebf1*) and fatty acid synthase (*Fasn*). Genes involved in lipid and fatty acid uptake [50] include very-low-density lipoprotein receptor (*Vldlr*), FAT atypical cadherin 1 (*CD36 = Fat*), and fatty acid transport proteins *Slc27a2 = Fatp2*, and *Slc27a5 = Fatp5*) [51]. Downregulated genes in SF mice could not be reversed using probiotic treatment. Significant differences between the groups are indicated. WTC n = 6; SFC n = 6; and SFL n = 5. * *p* < 0.05; ** *p* < 0.01; *** *p* < 0.001; **** *p* < 0.0001.

**Figure 4 nutrients-16-00511-f004:**
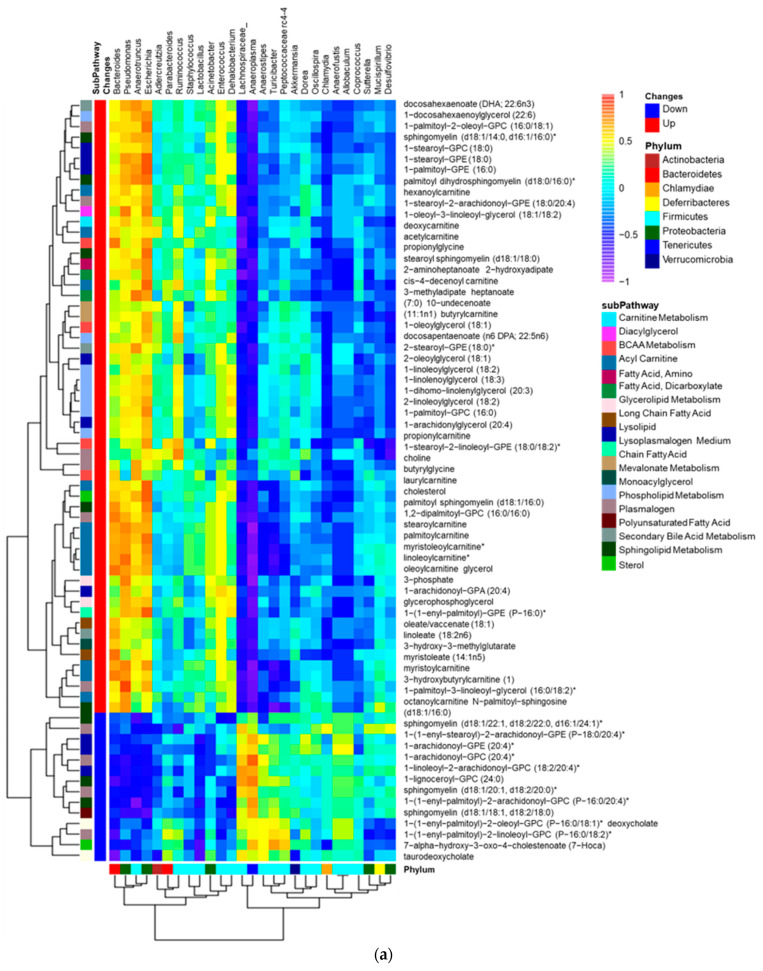
Correlation of gut microbiota and plasma lipids comparing SFC with WTC. (**a**) Correlation heatmap: bacterial genera (Labeled panel on the top) and lipid metabolites (labeled panel on the right), showing positive (orange/yellow) and negative (blue) correlations with changed lipids in SFC (n = 6) compared to WTC (n = 6). Upregulated (red) lipids and downregulated (blue) lipids are shown; note that most upregulated lipids in SF mice were positively correlated with *Pseudomonas*, *Anaerotruncus*, *Escherichia*, and *Bacteroides*. * indicates compounds that Metabolon Inc. has confident in its identify, however, compounds have not been officially confirmed based on a standard. (**b**) Dot graph indicating Spearman correlation coefficient Rho (*x*-axis) and significant correlation *p* value (<0.05, *y*-axis) between genera microbiota (the colors) and plasma lipid sub-pathways (the dot sizes) changed in SF mice compared to WT mice. Pearman’s Rho measures the strength of positive (the right) or negative (the left) association between genera microbiota and sub-pathways of lipids. Colors represent different genera of bacteria. Dot sizes represent different lipid sub-pathways. Twelve identified genera were significantly associated with 19 different lipid sub-pathways. Importantly, *Escherichia* (purple), *Pseudomonas* (orange), and *Anaerotruncus* (red) positively correlated significantly with altered levels of lipids, while *Anaeroplasma* (blue) and *Ruminococcus* (black) negatively correlated significantly with altered lipids in SF mice.

**Figure 5 nutrients-16-00511-f005:**
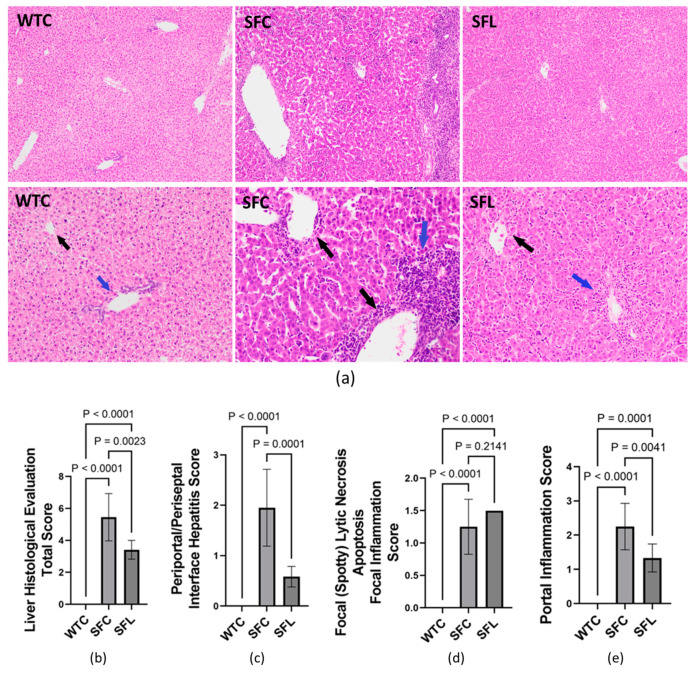
Hepatitis histological evaluation using the modified hepatic activity index (modified HAI) grading system. (**a**) Photomicrographs of hepatic tissue H&E-stained sections in WTC (left), SFC (middle), and SFL (right). The top panel shows 100× magnification, while the bottom panel shows 200× magnification. Compared to WT mice, SFC mice livers showed heavy portal tract and periportal chronic inflammation with interface hepatitis (blue arrow) and central and portal vein endotheliitis (black arrows). SFL mice with marked reduction of portal tract (blue arrow) and subendothelial central vein lymphocytic inflammation (black arrow). (**b**) Total histological score = periportal or periseptal interface hepatitis + confluent necrosis + focal inflammation + portal inflammation + architectural changes/fibrosis/cirrhosis). (**c**) Periportal/periseptal interface hepatitis score. (**d**) Focal lytic necrosis/apoptosis/focal inflammation. (**e**) Portal inflammation score. WTC (n = 6), SFC (n = 6), and SFL (n = 5); significant *p* values are indicated in the figures.

**Table 1 nutrients-16-00511-t001:** Lipid biochemicals associated with IFN-γ in SF mice.

Lipid Pathway	Lipid Biochemical	R Value	*p* Value
Lysolipid	1-linoleoyl-GPA (18:2)	0.94	0.015
Medium-Chain Fatty Acid	caprylate (8:0)	0.89	0.033
Sphingolipid metabolism	stearoyl sphingomyelin (d18:1/18:0)	0.89	0.033
Primary Bile Acid Metabolism	taurochenodeoxycholate	0.94	0.017
Secondary Bile Acid Metabolism	taurohyodeoxycholic acid	0.93	0.007

**Table 2 nutrients-16-00511-t002:** Lipid biochemicals associated with IL-4 in SF mice.

Lipid Pathway	Lipid Biochemical	R Value	*p* Value
Lysolipid	1-(1-enyl-oleoyl)-GPE (P-18:1)	0.94	0.017
	1-stearoyl-GPE (18:0)	0.89	0.033
	1-arachidonoyl-GPA (20:4)	0.99	0.0003
Phospholipid Metabolism	1-palmitoyl-2-oleoyl-GPC (16:0/18:1)	0.89	0.033
Long-Chain Fatty Acid	eicosenoate (20:1)	0.94	0.017
	erucate (22:1n9)	0.94	0.017
Polyunsaturated Fatty Acid	adrenate (22:4n6)	0.94	0.017
Lysoplasmalogen	1-(1-enyl-palmitoyl)-GPE (P-16:0)	0.94	0.017

## Data Availability

Most data generated or analyzed during this study are included in this published article. Those data generated and/or analyzed during the current study that are not published here are available from corresponding authors on reasonable request.

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
