# Peer review of "Probiotic *Limosilactobacillus reuteri* DSM 17938 Changes Foxp3 Deficiency-Induced Dyslipidemia and Chronic Hepatitis in Mice"

_nutrients, 2024, doi:10.3390/nu16040511_

Round 1
Reviewer 1 Report
Comments and Suggestions for Authors
General comments
The aim of this very interesting study was to determine the effect of a probiotic bacteria (Limosilactobaillus reuteri DSM 17938) on blood plasma dyslipidemia and liver morphological and biochemical changes caused by hepatitis. The authors used multiple state-of-the-art research methods including metabolomic profiling and gut microbiota determination using the Next Generation Sequencing (NGS) analysis performed on unique animal model (the scurfy mouse with immune dysregulation, enterpathy, like human IPEX syndrome linked to various autoimmune disorders).
Taking into consideration the probiotic bacteria with potentially huge therapeutic action, the unique animal model and the research techniques used, both the Introduction and Discussion chapters are very disappointing.
Introduction: which should be a brief overview of the current state of knowledge, is mostly based by the authors on very old references (e.g. 1991 or 1999). The entire chapter needs to be rewritten using the latest references. It should be added that there is a significant contrast between the two parts of the Introduction (the part to be completely rewritten lines 39 to 48, the other part which presents the current state of knowledge from lines 49 to 65). The first part of Introduction should to be rewritten. Furthermore, the last sentence of the Introduction should have a follow-up.
Materials and Methods: many details are missing from the description of the in vivo experiment, including environmental parameters (temperaturę, humidity, air change ratio, photoperiod) and characteristics of the feed (pelets) that the mice consumed. The use of the term 'orally' for gavage administration is incorrect, it is an intragastric administration, it should be changed throughout the all manuscript, including the description of the results.
Results: illustrations showing the profile of the microbiota should be enriched with more figures, including, for example, Chao1 index, Pielou’s evenness index, Shannon diversity. Also, histopathology results should have more detailed photographic documentation.
Discussion: the discussion of such interesting results is too cursory, failing to explain the potential mechanisms that accompanied the probiotic effect found. Some of the references should be more recent and not a dozen years old. This part of the manuscript should be rewritten.
Conclusions: these are observations from the study rather than conclusions from the results. Should be rewritten
Abstract: line 30-33 to be rewritten, this sentence is too speculative.
Other comments:
Line 92 – unclear definition of d22, correct
Line 196 – delete or replace with newer old reference (# 33)
Line 356 – the information regarding the animal model should be clarified - guinea pig is not pig (domestic, Sus scrofa L.). Fundamental difference
Line 367 – speculation too far, this has not been investigated
Line 385 – please provide newer references
Author Response
Please see attached pdf with responses to reviewer 1.

Reviewer 2 Report
Comments and Suggestions for Authors
The manuscript entitled - Probiotic Limosilactobacillus reuteri DSM 17938 Changes 2 Foxp3 Deficiency-induced Dyslipidemia and Chronic Hepatitis 3 in Mice by Kostandy EN et al., is an interesting work and needs further clarity. Some of the points to be addressed are described below;
1 . The statistical analysis part needs to be elaborate and should be clearly explained.
2 . In the figure legends for all figures, it needs to be mentioned how many replicates were used.
3 . Figure 5 a has only disease condition, which is SFC group, it is important and must to include WTC and SFL groups with histological evaluation.
4 . Necrotic markers detection using Western blot or Immunoflourescence should be performed in the hepatic tissue sections.
5 . AST and ALT levels should be measured in all the treatment groups and compared.
6 . Include a limitations section before the conclusion part.
Comments on the Quality of English LanguageMinor edits and proofreads are required.
Author Response
PLease see attached responses to reviewer 2.

Reviewer 3 Report
Comments and Suggestions for Authors
The present manuscript titled ‘Probiotic Limosilactobacillus reuteri DSM 17938 Changes Foxp3 Deficiency-induced Dyslipidemia and Chronic Hepatitis in Mice’ is interesting Topic and written well. However, some comments need to pay attention to it.
Comment 1: The introduction provides a clear background on the scurfy mouse model and its relevance to IPEX syndrome. However, it would benefit from a more concise presentation. Consider expanding the introduction since it is quite short for improving the readability. And clarify the hypothesis at the end of the introduction.
Comment 2: Clarify the significance of the age range chosen for the mice (6- or 8-week-old). Does the age selection relate to a specific stage in disease progression?
Comment 3: The details on DSM 17938 preparation and administration are well-documented. However, it might be helpful to include information on dosage rationale or any existing literature supporting the chosen administration method.
Comment 4: The discussion on the potential role of PPARα is intriguing. Further exploration of the implications of PPARα downregulation in the context of autoimmune diseases and the impact of probiotics on PPARα-related pathways would enhance the discussion.
Comment 5: The conclusion could be strengthened by summarizing the key findings and their potential implications. Consider explicitly stating the novelty of the study and potential future research directions.
Overall Comments:
The manuscript is well-written and contains valuable scientific insights. However, improving the organization of certain sections and providing a more concise presentation in the introduction would enhance the overall readability.
Author Response
Please see attached pdf with responses to reviewer 3

Round 2
Reviewer 1 Report
Comments and Suggestions for Authors
Thank you very much for taking my comments into account, in the current version I consider the manuscript to meet all the requirements for this type of work
Reviewer 2 Report
Comments and Suggestions for Authors
Please wait for the response from the Editorial office.
Reviewer 3 Report
Comments and Suggestions for Authors
No more comments